**Using Satellite Measurements of $N_2O$ to remove dynamical variability from HCl measurements**

Richard S. Stolarski
Johns Hopkins University

Anne R. Douglass, Susan E. Strahan
NASA Goddard Space Flight Center

*Abstract:*

Column HCl measurements show deviations from the expected slow decline following the regulation of chlorine-containing compounds by the Montreal Protocol. We use the simultaneous measurements of $N_2O$ and HCl by the MLS instrument on the Aura satellite to examine this problem. We find that the use of $N_2O$ measurements at a specific altitude to represent the impact of dynamical variability on HCl results in a derived linear trend in HCl that is negative (ranging from -2.5%/decade to 5.3%/decade) at all altitudes between 68 hPa and 10 hPa. These trends are at or near $2\sigma$ statistical significance at all pressure levels between 68 hPa and 10 hPa. This shows that analysis of simultaneous measurements of several constituents is a useful approach to identify small trends from data records that are strongly influenced by dynamical interannual variability.

**I. Introduction**

HCl is the primary constituent of inorganic chlorine in the stratosphere, comprising 75-80% of the inorganic chlorine in the vertical pressure range from 68 hPa to 10 hPa [Zander et al., 1992; Nassar, et al. 2006]. As such it provides a convenient marker for the total amount of inorganic stratospheric chlorine. This marker can be measured from the ground as a total column amount and from satellites as a vertical profile. The column amount of HCl is expected to follow the behavior of the concentration of the organic sources of chlorine as measured at the surface (e.g. CFCs) with a time delay of a few years for the CFCs to reach the stratosphere where they are converted to inorganic chlorine compounds.

 Rinsland et al. [2003], using the NDACC record of ground-based column measurements of HCl and $ClONO_2$ ($\sim$1990 – 2002), showed that their total stratospheric burden had leveled out by approximately 1995. These two gases comprise most of stratospheric inorganic chlorine outside the winter polar vortices. The next step would be to observe the expected decrease in inorganic chlorine. Recently Mahieu et al. [2014] have shown that, in fact, the measured HCl column over Jungfraujoch decreased more rapidly than expected from in-situ ground-based measurements of source gases during the early 2000's. The rapid decrease in measured HCl column was followed by an increase from about 2007 to 2010 even as the chlorine gases were decreasing. We show, as Mahieu et al. [2014] indicated,

that the Microwave Limb Sounder (MLS) measurements of the lower stratospheric
column of HCl also decrease and increase in concert with the Jungfraujoch data,
followed by a decrease from 2011 to mid 2013 and a subsequent increase from that
time to the present (see Figure 1).

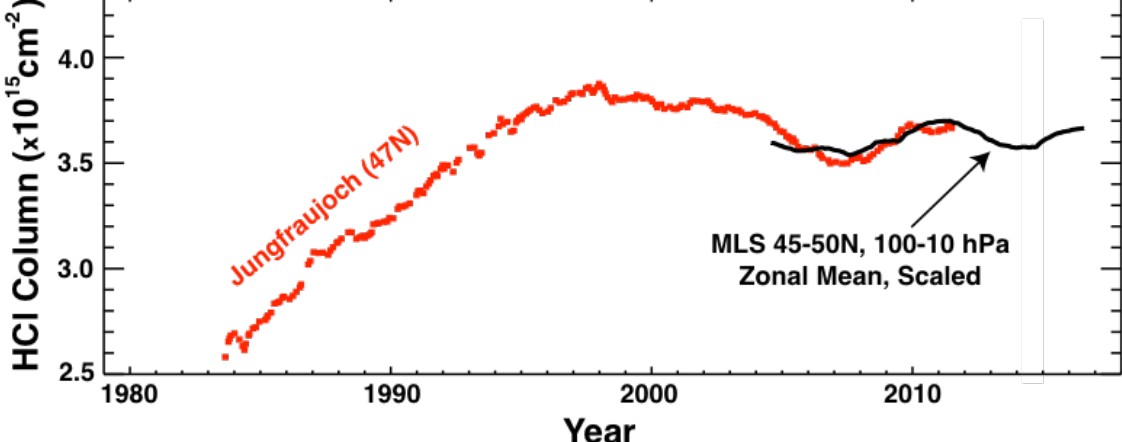

Figure 1: Total column measurements of HCl above Jungfraujoch (latitude = 46.5°N)
smoothed with a 3-year running mean as shown by Mahieu et al. [2014] (red curve).  Also
shown are the 3-year running mean smoothed zonal mean of measurements of the lower
stratospheric column of HCl (100-10 hPa) from the MLS instrument on Aura for the latitude
band from 45 to 50N (black curve).  The MLS measurements are of partial column and have
been scaled upward to match the Jungfraujoch FTIR measurements for better visual
comparison.
Mahieu et al. [2014] use results from model simulations with the SLIMCAT model
driven by ERA-Interim meteorological fields from the European Centre for Medium-
Range Weather Forecasts (ECMWF) to suggest that variability in the stratospheric
circulation causes the accelerated decrease and the unexpected increase in HCl
column. We will explore this explanation using measurements of $N_2O$ from MLS as a
measure of this variability in circulation.
**2. MLS Data: HCl and $N_2O$**
We use the MLS HCl and $N_2O$ data together to test whether chlorine is decreasing in
the stratosphere as expected from adherence to the provisions of the Montreal
Protocol.  MLS was launched on the Aura Satellite in July 2004 [Waters et al., 2006]
and continues to operate in 2018.  The record is now more than 13 years in length
with vertical profiles of HCl, $HNO_3$, $N_2O$ and many other species measured globally
on a daily basis.
For HCl we use the version 4.2 product, measured by the 640 GHz receiver, that
shows little change from the previous version 3 products.  According to the MLS
data quality document [Livesey et al., 2017] the useful range for HCl measurements
is from 100 to 0.32 hPa.  However, the useful data for trends are limited to pressures
greater than 10 hPa due to the insufficient reliability of the retrievals in the upper
stratosphere.  The possibility of a temporal drift in the HCl product was evaluated by
comparing the drift between the $O_3$-240 and $O_3$-640 (L. Froidevaux, personal
communication, 2017). Because the ozone products had a drift of ≤0.1%/yr, other
measurements obtained with the 640 GHz receiver, e.g., HCl and $N_2O$, are expected
to be comparably stable. An evaluation of the $O_3$-240 GHz product compared to
correlative satellite and ground-based measurements shows no evidence of a
temporal drift (Hubert et al., 2016).
For $N_2O$ we use the version 4.2 data product from 190 GHz receiver because the $N_2O$
640 GHz data set ends in summer 2013 due to failure of the $N_2O$ primary band. The
MLS measurements with the 190 GHz receiver have been found to have a temporal
drift relative to the 640 GHz receiver (N. Livesey, personal communication, 2017).
Later in this section we will compare $N_2O$-190 and $N_2O$-640 measurements to
correct for the drift observed between these two channels prior to 2013.  The 190-
GHz $N_2O$ data are stated to be useful in the 68-0.46 hPa range.  We will thus restrict
our analyses to pressure levels between 68 and 10 hPa in this paper where both the
HCl and $N_2O$-190 measurements are useful.

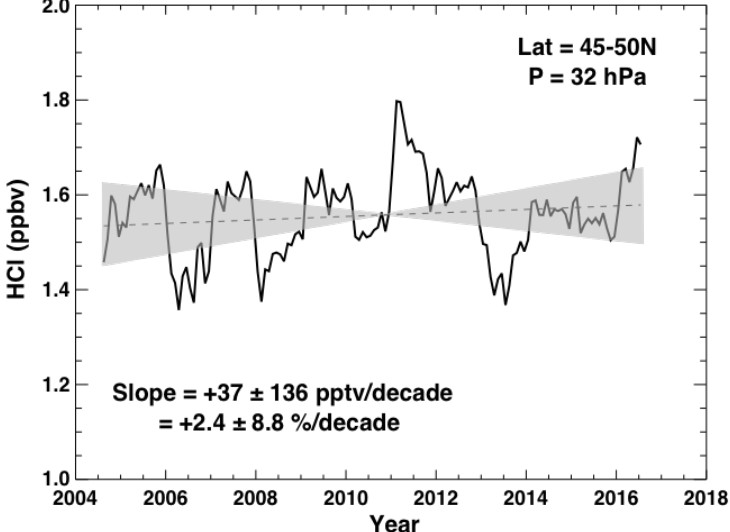

Figure 2: Deseasonalized monthly-mean MLS measurements of HCl concentration at 32 hPa.
Measurements are area-weighted between 45°N and 50°N.  The dashed line is a linear least
squares fit to the data and the shaded area indicates the 2σ uncertainty in that fit including
consideration of auto-correlation in the time series.
To better understand the variations in HCl column amounts at 47°N observed by
Mahieu et al. [2014] we begin with consideration of the MLS measurements of the
HCl profile at specific pressure levels in the stratosphere.  For example, Figure 2
shows the deseasonalized monthly mean measurements by MLS of HCl averaged
between 45°N and 50°N latitude at 32 hPa.  The data are shown as mixing ratio after
removal of the repeating seasonal cycle.  The data clearly show deviations of as
much as ±10% with significant auto-correlation.
One way to examine the HCl time series shown in Figure 2 is to attempt to "explain"
the variance by fitting to various measures of dynamical variability such as the
Quasi-Biennial Oscillation (QBO) or El-Niño/Southern Oscillation (ENSO).  This
method may remove much of the dynamical variance but has at least two potential
problems: 1) the fitting parameters may only remove part of the dynamical
variability because of incomplete representation of that variability and 2) they may
over-represent the variability because of correlation between parameters.  Either of
these problems could lead to difficulties in separating real trends from apparent
trends in the residual over short time scales such as the 12 years of data since 2004.
For example, the impact of the QBO on southern mid-latitude composition depends
on the QBO phase during early (southern) winter [Strahan et al. 2015].  The
resulting dynamical variability is not easily represented by fitting a QBO plus a
seasonal term in a statistical model, because the actual variability depends on the
QBO phase during a particular season.  We have attempted to model the HCl time
series with standard proxies for QBO and other effects with unsatisfactory results.
We use a different method to remove dynamical variability in the HCl data set,
taking advantage of simultaneous measurements of another species made by the
MLS instrument on the Aura satellite.  The observed deviations from the seasonal
variation of HCl and other constituents including $N_2O$ are the result of dynamical
variability acting on mixing ratio gradients.  These gradients may be vertical,
horizontal, or a combination of both.  If two constituents have gradients in the same
or opposite directions, the impact of dynamic variability will be to cause deviations
that are either correlated or anti-correlated with each other depending on the sign
of the gradients.  An example is shown in Figure 3 where we plot the deseasonalized
HCl mixing ratios at 32 hPa for the latitude band 45-50N as in Figure 2 and the
deseasonalized $N_2O$ mixing ratios on a reverse scale for the same latitude band and
pressure level.  The correlation coefficient is -0.87 between these two time series.  A
similar correlation, with opposite sign, is found between HCl and $HNO_3$ data from
MLS.

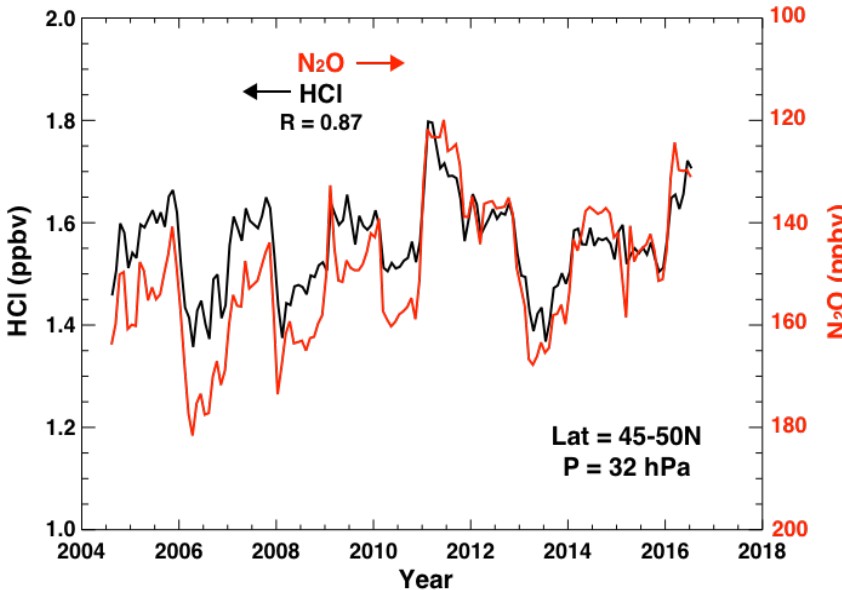

Figure 3: Deseasonalized time series of MLS HCl measurements (black curve, same as Figure
2) and deseasonalized time series of MLS $N_2O$ measurements plotted with reverse scale on
right side of figure(red curve) for the latitude band 45-50N at 32 hPa pressure level.
**3. Time Series Analysis: Using $N_2O$ Measurements as a Fitting Parameter**
The trend that we are trying to isolate and confirm for the HCl time series is
determined by the change in abundance of chlorine-containing halocarbons driven
by the provisions of the Montreal Protocol on ozone-depleting substances.
Inorganic chlorine in the stratosphere is expected to have decreased since 2000 in
response to the decreases in the chlorine-containing source gases. $N_2O$, on the other
hand, is known to be increasing at a rate of about 2.8%/decade [NOAA GMDL data
updated from Elkins and Dutton [2009] available at
ftp://ftp.cmdl.noaa.gov/hats/n2o/combined/HATS_global_N2O.txt]. Our approach
is to use the $N_2O$ time series at each altitude, such as that shown in Figure 3, as an
explanatory variable in a time-series regression to remove the dynamical variability
from the HCl time series. Trends calculated for HCl in this time-series regression
are then corrected for the underlying trend in $N_2O$.
The time series method used is a simple regression with two fitting terms, a linear
trend and the $N_2O$ time series. Uncertainties are estimated by calculating the
standard deviation of the residual time series after removing the fit to linear trend
plus the $N_2O$ time-series and then multiplying the result by the factor
$((1+\Phi)/(1-\Phi))^{1/2}$ to obtain the uncertainty including auto-regression (see
Weatherhead et al. [1998]) where $\Phi$ is the auto-correlation lag 1 coefficient.

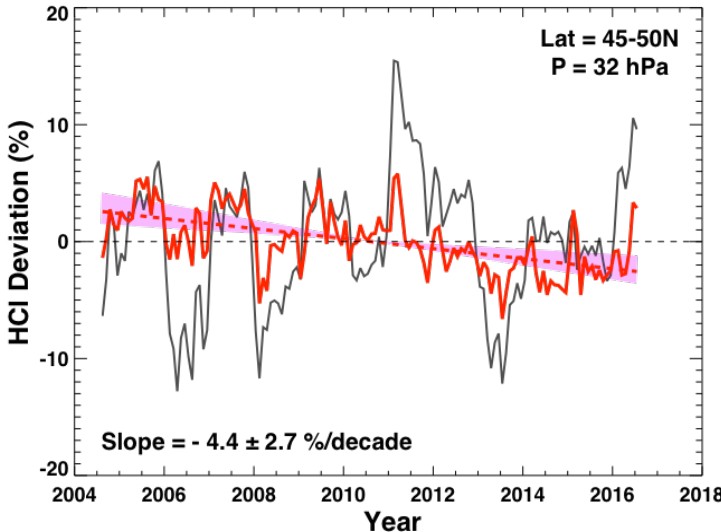

Figure 4: HCl anomaly time series as in Figures 2 and 3 with the mean removed (black) and the residual time series after regression to $N_2O$ time series (red). Red dashed line is linear fit to residual series with $2\sigma$ uncertainty bounds indicated by shaded area.

The result of using $N_2O$ as a fitting parameter for the 32 hPa MLS time series for HCl is shown in Figure 4. The solid red line in the figure is the residual time series after fitting, which takes advantage of the substantial covariance and shows significantly reduced variability. The resulting trend shown by the red dashed line is $-4.4 \pm 2.7$ ($2\sigma$) %/decade. The HCl trend at this pressure level is now negative and statistically significant at more than the $3\sigma$ level. The same procedure has been carried out at each of the pressure levels for MLS retrievals. The result is shown the third column of Table 1.

The second column of Table 1 shows the raw trend obtained from the MLS HCl measurements. We can see that the raw trend is essentially the same as the $N_2O$-fitted trend at the two highest levels (10 and 15 hPa) where inorganic chlorine is relatively insensitive to dynamical variability due to its small vertical gradient. At higher pressures, where inorganic chlorine has a larger vertical gradient, the trend is substantially different when using $N_2O$ as a fitting parameter. Using $N_2O$ as a fitting parameter changes the lower level trends from positive to negative with a substantial reduction in the uncertainty.

A further problem alluded to earlier in this section is that MLS now uses the 190-GHz band for its standard $N_2O$ product because the 640-GHz band is no longer usable. The problem arises because the 190-GHz band displays a drift in $N_2O$ measurements with respect to the 640-GHz band [L. Froidevaux, pers. commun.]. Since the 640-GHz band was found to be stable during its operational period, we use the drift rate of the 190-GHz band with respect to the 640-GHz band calculated by the time-independent drift rates for the time period in which they both were operational (2004-2012). We assume that the drift rate computed over the overlap

time period continues through the end of the data record.  This assumption is
subject to an uncertainty that we have no way of evaluating.  The results for the
computation of the drift between the 190-GHz band with respect to the 640-GHz
band are shown in the fourth column of Table 1 with estimated uncertainties in the
fit.
We chose to show the trend resulting from the fit to the $N_2O$ product before
adjusting it for the drift between the two $N_2O$ bands and then following that with a
correction for the $N_2O$ drift.  Because we are making a simple straight-line trend
correction with no time variability except the trend, the result is the same as that
obtained by first correcting the $N_2O$ product time series and using that result as a
fitting parameter to the HCl data.

| Pressure Level (hPa) | Raw HCl Trend (%/dec) | Trend with $N_2O$ fit (%/dec) | $N_2O$ 190/640 drift (%/dec) | Surf $N_2O$ Trend (%/dec) | Final HCl Trend (%/dec) |
|---|---|---|---|---|---|
| 10 | -3.7±2.0 | -3.0±2.0 | -5.1±3.7 | +2.8±0.05 | -5.3±4.2 |
| 15 | -3.8±3.1 | -3.5±1.9 | -2.9±2.2 | +2.8±0.05 | -3.6±2.9 |
| 22 | -1.3±4.2 | -3.8±2.3 | -1.5±1.3 | +2.8±0.05 | -2.5±2.6 |
| 32 | +2,4±8.8 | -4.4±2.7 | -2.2±1.4 | +2.8±0.05 | -3.8±3.0 |
| 46 | +3.8±7.2 | -2.3±2.3 | -5.2±1.8 | +2.8±0.05 | -4.7±2.9 |
| 68 | +3.9±4.0 | -1.3±2.7 | -5.0±1.6 | +2.8±0.05 | -3.5±3.1 |

Table 1. HCl trends derived from MLS data at 6 pressure levels.  Column 1 gives the
pressure level.  Column 2 gives the raw trend derived directly from the MLS HCl
measurements in %/decade.  Column 3 gives the HCl trend derived using MLS $N_2O$
measurements as an explanatory variable.  Column 4 gives the derived trend in the 190-GHz
channel of MLS $N_2O$ measurements relative to the 640-GHz channel during the time of their
overlap.  Column 5 gives the slope of the NOAA global surface measurements of $N_2O$ over
the period from 2001 to 2012.  Column 6 gives the overall resulting trend obtained by
combining the information in columns 3 to 5.  All uncertainties are quoted at $2\sigma$.
Finally, we note that the surface levels of $N_2O$ increased by about 2.8%/decade.  We
used the "global" nitrous oxide data reported at the NOAA ESRL Global Monitoring
Division web site (https://www.esrl.noaa.gov/gmd/hats/combined/N2O.html).  We
fit a trend to the data between 2001 and 2012 to represent the ground values that
should be seen in the stratosphere about 3 years later in the MLS data.  The time
delay represents the approximate mean age of air in the lower stratosphere (see e.g.
Waugh and Hall [2002].  Since $N_2O$ has been increasing at a nearly constant rate, the
choice of time period for estimating its trend does not lead to a significant
uncertainty.
The final result is obtained by summing the trends in columns 3 to 5 of Table 1 and
is shown in column 6 of the table.  The uncertainties were obtained by using the root
sum of squares (RSS) of the uncertainties in columns 3 to 5.  This final result is also
shown in Figure 5.  The blue solid line indicates the trend result obtained from a
linear fit to the deseasonalized residuals with no attempt to account for dynamical
variability (column 2 of Table 1).  The red solid curve is the result when the $N_2O$
time series is used as a dynamical surrogate, with corrections for drift and surface
trends, in the fitting procedure (column 6 of Table 1).  The shaded areas represent
$2\sigma$ uncertainties in the linear trends.
Note in Figure 5 that the use of $N_2O$ as a surrogate for dynamical variability reduces
the uncertainty in the calculated trend for all of the pressure levels between 15 hPa
and 68 hPa where the MLS data for both HCl and $N_2O$ are considered to be suitable
for trend analysis.  Figure 5 also shows that the calculated linear trend in HCl using
the $N_2O$ surrogate with corrections is negative at all pressure levels and is
significantly negative ($2\sigma$) at all levels from 68 hPa to 10 hPa with the exception of
22 hPa where it has nearly $2\sigma$ significance.  These results are consistent with the
observed decrease in organic chlorine species at the surface.

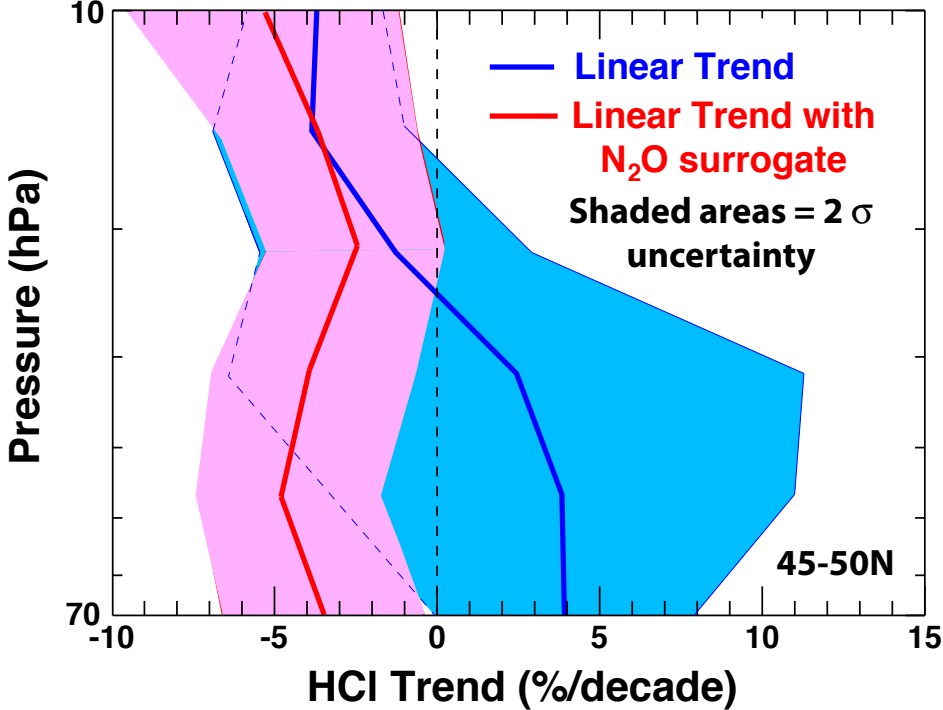

Figure 5: Linear trend in HCl concentrations determined from MLS measurements between
70 and 10 hPa (approximately 20 to 30 km altitude) for the latitude band of 45-50N.  The
blue line is the trend determined from the raw deseasonalized data.  The red curve is the
trend determined while including the $N_2O$ time series as an explanatory variable.  The
shaded areas represent $2\sigma$ uncertainties for each.
**4. Conclusion**

In their paper, Mahieu et al. [2014] reported that total column HCl measured at Jungfraujoch showed significant variation from the expected simple linear decrease. We have attempted to quantitatively evaluate the effect of dynamical variability on the concentrations of HCl in the stratosphere by using $N_2O$ measurements since late 2004 made by the Aura MLS instrument. Since both HCl and $N_2O$ have concentration gradients (horizontal and vertical) that are acted upon by dynamical processes to create inter-annual variability, we have used the variability of $N_2O$ concentrations determined from MLS observations as a measure of the dynamical variability that should be expected in HCl concentrations. We suggest that this method more reliably removes the real atmospheric variability than does the use of other proxies.

We have shown that using an $N_2O$ surrogate in trend analysis of the MLS HCl time series results in a trend that is negative at all measured levels from 68 hPa upward to 10 hPa and that these negative trends are $2\sigma$ statistically significant or nearly so. The $N_2O$ surrogate had little effect at the upper two levels of 10 and 15 hPa where inorganic chlorine is less sensitive to dynamical variability. The surrogate had significant impact on the derived trends lower in the stratosphere where the inorganic chlorine vertical gradient is larger and most of the HCl column resides.

Previous data-based estimates of HCl decrease in the stratosphere include Froidevaux et al. [2006], Jones et al. [2011], Brown et al. [2011], and Kohlhepp et al. [2012]. All of these studies had to consider the issues we have discussed in this paper, namely the contribution of dynamic variability to the apparent trend. In each case, the shortness of the data record was a significant limitation to the interpretation of potential trends due to the decrease in tropospheric organic chlorine sources.

The results from these authors are summarized in the 2014 Ozone Assessment Report [Carpenter and Reimann, 2014]. Specifically Froidevaux et al. [2006] derived a trend for the 50-65 km altitude range of -8%/decade from MLS data for the years 2004-2006. Jones et al. [2011] derived a trend of -5%/decade using HALOE and ACE FTS data between 35 and 45 km from 1997 to 2008 at midlatitudes. Brown et al. [2011] deduced a trend of -7%/decade for the 50 to 54 km range from ACE FTS data from 2004 to 2010. Finally, Kohlhepp et al. [2012] analyzed the total column HCl data from 17 NDACC FTIR stations for the years 2000 to 2009, obtaining trends that ranged from -4 to -16%/decade depending on station.

The best comparison for evaluating our results is considering the change in the organic chlorine sources at the surface. The 2014 ozone assessment, Chapter 1, [Carpenter and Reimann, 2015] estimates changes in the tropospheric available organic chlorine of -6%/decade from 2000-2004 followed by -4.6%/decade from 2004-2008 and -4%/decade from 2008-2012. Assuming a 3-5 year delay between changes in the tropospheric source gases for the stratospheric chlorine implies an

average change from 2004 to 2016 of about -5 %/decade, in agreement with our estimate from MLS data within the uncertainty bounds.

These results indicate the potential power of using the time series of measurements of one constituent to understand, and possibly remove, the dynamical variability in another constituent.  In the case we have presented, we had to apply a drift correction to the MLS $N_2O$ data to get the best estimate of trend.  We have attempted to make an estimate of the uncertainty in this drift correction and include it in the estimate of the overall uncertainty in the trend calculation.  Although the drift correction was a large enough fraction of the overall trend to cause some worry about the results, we assert that this does not diminish the value of the concept of using the dynamical variability of a other measured constituents to provide complementary information about trends and variability of the constituent whose possible trends are under consideration.

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

        186.