# Peer review of "Using Satellite Measurements of N2O to remove dynamical variability from HCl measurements 3 Richard S. Stolarski 4 5 Johns Hopkins University 6 7 Anne R. Douglass, Susan E. Strahan 8 NASA Goddard Space Flight Center 9 10 Abstract: 11 12 Column HCl measurements show deviations from the"

_Atmospheric Chemistry and Physics, 2017_

## Referee Comment (RC1) · Anonymous Referee #1 · 10 Jan 2018

This manuscript introduces a method aiming at accounting for (correcting for) the real variability observed in time series of stratospheric tracers. More specifically, simultaneous measurements of hydrogen chloride (HCl, the main reservoir of stratospheric chlorine) and nitrous oxide (a long-lived source of nitrogen) are used in conjunction, so that the variability of stratospheric N2O is used to remove the one of HCl, assuming they have the same origins, here circulation and transport. The ultimate aim is to determine HCl trends unaffected by atmospheric variability, useful for the verification of the effectiveness of the Montreal Protocol in the stratosphere.

It is claimed that this method is more appropriate than those using proxies for representing multiyear dynamical variabilities resulting from, e.g., the QBO, ENSO. In the present case, the implementation of the method to simultaneous measurements of HCl and N2O by the Aura/MLS instrument results in the determination of significant HCl decreases in the lower stratosphere over the 2004-2016 time period (about 13 years) which are in agreement with the evolution of total organic chlorine at the surface, when accounting for the mean age of stratospheric air. In contrast, direct HCl trends are mostly not significant over this decadal time scale.

One can foresee that the method presented in this study will be used in future trend evaluations, for HCl, but also for other target gases relevant to ozone depletion and recovery, when trying to reconcile tropospheric and stratospheric trends,. . . provided that simultaneous measurements of tracers with similar response to atmospheric dynamical variability or other influences are available.

Therefore, I recommend publication of this study, after consideration of the suggestions indicated below.

Major comments

In order to allow precise implementations of this approach or replication of the method in future studies, it would be good to have available a description on how the "time-series regression" is actually performed (by simple ratio-ing or more elaborated ways). I believe this would not be possible with the current version of the text.

Another aspect which is not described is the evaluation of the uncertainties affecting the various trends. It is stated in caption of Fig. 2 that "the $2\sigma$ uncertainty in that fit includes consideration of auto-correlation in the time series". Various approaches have been used in recent papers such as to account for auto-correlation in the data sets, particularly for studies of ozone recovery. But we do not know how the uncertainty ranges were determined in the present instance, while visual inspection of the HCl and N2O time series suggests that auto-correlation might be quite significant. It would be good to know how the authors accounted for auto-correlation. A brief description of the

statistical evaluation of the confidence intervals should be added. A citation might be relevant if the current method has been used before.

Specific comments and remarks

There is a mismatch between the main text and the captions for Fig. 2 and 3. It is indicated in the text that anomalies or deviations are shown. But it looks like the captions correctly state that deseasonalized time series are shown for HCl and N2O. Several portions of the main body text need to be amended accordingly, or the captions and figures 2 and 3 updated.

Minor comments or typos

-Page 1/line 30: suggest changing to "... a few years for them to reach..."

-Page 1/line 37: Jungfraujoch is misspelt

-Page 1/line 38: suggest changing to "...from in situ surface measurements..."

-Page 2/line 14: might be good to indicate why N2O is a relevant target for this purpose (and/or add a reference)

-Page 3/line 1: I don't think that there is a comparison between the N2O products from MLS, only the drift as a function of altitude is given

-Page 4/line 24: "deseasonalized" is misspelt

-Page 5/line3-line5: this sentence needs to be reworded

-Page 5/line 16: "HCl anomaly time series as in Figures 2 and 3"; true if Fig. 2 and 3 are updated accordingly

-Page 6/line28-30: a good reference is needed here, to introduce the concept of age of air, and showing that a 3 years delay is appropriate

-Page 7/figure 5: perhaps provide approximate altitude information on the right scale?

-Page 7/line 23: suggest replacing "showed" by 'reported"

-Page 7/line 24: suggest replacing "linear trend" by "linear decrease"

-Page 8/line 36: [Carpenter and Reimann, 2014] instead of 2015

-Page 9/line 8: 2014 instead of 2015

---

## Referee Comment (RC2) · Anonymous Referee #2 · 17 Jan 2018

General overview

This paper presents a nice perspective on observed variability and trends in northern mid-latitude stratospheric HCl. It describes an approach whereby dynamical influences on that variability can be accounted for through consideration of a trace gas such as N2O, which shares many of the dynamical influences as HCl but experiences different chemical processes.

While I recognize the value of the method described, and am keen to see it appear in the literature, I am concerned the uncertainties ascribed to some of the numbers found/used by the authors are on the optimistic side. I wonder if more complete as-
sessment of these uncertainties might lead to a reduction in the reported "significance" of the result and this assessment might then suggest that a softening of some of the wording is merited. I also have a concern as to whether the ordering of the operations in their method is appropriate, and whether more robust results might be obtained if it were reversed. Both of these topics are expanded upon below.

The standard of English is reasonably high, but it would clearly have benefited from a more careful read through by the authors as there are several parts that are erroneously and/or ambiguously worded. I've endeavored to identify some of these, but fear I may have overlooked some others.

Major concerns

My concern about the ordering of steps in the method is as follows. I would have thought that it would have been better to "correct" the N2O for both the likely MLS drift and the surface growth rate before using it as an explanatory variable in the HCl analysis rather than, as appears to be the case, after. My sense is that this would lead to a corrected N2O variable that would do a better job of explaining the dynamical influences on the HCl, enabling a clearer trend to be obtained. The results may be little different in the end, but my sense is that the study would be better expressed in that manner. If nothing else, the authors would do well to enact that alternative formulation and comment on the difference it makes to the result (even if they chose not to show it in the end). It might make sense to include an actual algebraic expression for the fit and the various corrections. This would make for an easier description for the various terms involved and their uncertainties.

My more major concern relates to the uncertainties quoted for some of the results. This is particularly important given the extent to which many of them are only just statistically significant (using the authors' 2-sigma threshold). Firstly, it is clear that the level-to-level variations in the bottom line results are mostly driven by the reported N2O 190/640 drift (r=0.75 between it and the result) rather than by the observed HCl trend (r=-0.24). That

is to say, the results are affected more by the "correction" than by the actual input (the latter being the HCl trend with the N2O fit term included). Accordingly, this correction deserves particular scrutiny. The degree of level-to-level changes in this drift term is large compared to the uncertainty quoted on many of the individual drifts. Arguably, the standard deviation (1-sigma=1.5%/decade) of these different estimates would be just as valid a measure of the uncertainty in any or all of them. Indeed it might have been just as valid to chose to use the multi-level-mean drift as the value for all levels, given the uncertainty introduced by the inherent assumptions being made. Foremost among those assumptions is the one that the N2O drifts seen in the first part of the MLS mission are the same as those expected in the post-2013 period, when the 640 GHz N2O product is unavailable. I would have thought that the uncertainties derived here might need to be inflated in some way to account for this. Might more information be gained through consideration of other MLS products measured in the same period? Fundamentally, I think more information is needed here (including from the MLS team) on these uncertainties and their validity.

My second concern on the uncertainty relates to the 0.05%/decade (2-sigma) uncertainty quoted on the impact of N2O emissions. Firstly, the use of a constant 2.8%/decade trend at all altitudes here strikes me as highly simplistic. There are factors such as changes in age of air (and its spectrum) that surely come into play and might lead to variations. Similarly, the use of a 3-year lag at all altitudes seems overly simplistic. I grant that these issues may only have a small impact, and they may be very hard to quantify from the measurements available. Thus, the use of a constant value may well be justified in that light. However, I find it hard to believe that, in the face of those issues, the 0.05%/decade 2-sigma uncertainty estimate is an appropriate one. If nothing else, I would urge the authors to validate this number through, for example, examination of CCM runs (to which this team has ready access). Quantifying the degree to which the modeled 45N N2O timeseries at different pressure levels tracks the surface trend would provide a useful measure of this uncertainty. This issue is perhaps tied up with the ordering one discussed above, as the use of N2O as an

explanatory variable for the sought-after HCl trends may absorb these factors to some extent (though I haven't thought this through fully). In my mind all these issues argue that a more complete exploration of their methods, their inherent assumptions, and the uncertainties therein should be included in the manuscript.

More minor points

— Page 1

Line 19: "Statistically" -> "Statistical"

Line 25: "altitude" -> "vertical" (as you're using pressures rather than altitudes in words that follow).

Line 27: "... amount of inorganic stratospheric chlorine. This marker can be ..." to avoid the ambiguity about whether it is the HCl or the inorganic chlorine that "can be measured from the ground and from satellites".

Line 33: Commas needed after "showed" and "measurements"

Line 34: "Inorganic chlorine" is more than just HCl and ClONO2, though granted the others may be minor. Or is the point that Rinsland et al. only measured those two species and argued that they are the bulk inorganic chlorine. Please clarify.

Line 37: Jungfraujoch misspelt

Line 38: "during the early 2000s. This was followed by an increase in the HCl column over Jungfraujoch from ..." to avoid the ambiguity about whether it is the HCl or the source gases (the most recent things being discussed) being referred to.

Lines 39-43: The way this is worded, it seemingly ignores the fact that Mahieu et al. also looked at this signal in MLS data (as embodied in the GOZCARDS dataset). Please reword accordingly.

— Page 2

Line 3: Quote the latitude of Jungfraujoch in the caption. Also, some redundancy, as you say the MLS data is a 100-10hPa column in one sentence and then talk about it being a partial column (without the numbers) later on.

Lines 10-15: Again, please be sure your wording is consistent with the use to which Mahieu et al. put MLS data.

Line 10: "results from simulations using the SLIMCAT model driven by..."

Line 21: July 2004 doesn't sound like "late 2004" to me.

Line 23: "altitude" -> "vertically resolved", given that the vertical coordinate is pressure.

Line 26: "has little change since" -> "shows little change from"

Line 30: Perhaps put "band 14" in quotes as it's jargon that's not explained earlier (and is presumably covered in the references given earlier in the paragraph).

Lines 32-34: Please clarify, has the N2O product been "redefined" since the release of v4.2, or was the redefinition part of v4.2 from the outset?

Line 34: Unless I've misunderstood, it's part of MLS that has "deteriorated" is it not? Starting at some point during the mission. The way this is worded it sounds like the MLS data files are somehow deteriorating with time (like food going off in the refrigerator) regardless of the time at which the observations were made. Please reword more precisely.

— Page 3

Line 1: "next" -> "following" sounds better to me.

Line 2: Are the "640 channel" measurements also from the v4.2 dataset or from some earlier version?

Lines 1-6: This would presumably be a good place to have a discussion about the validity of assuming that the pre-2013 drifts are representative of the post-2013 observations. (Or possibly on page 6, see later).

Figure 2: The way you've drawn this, with the shaded envelope being narrow at the left hand edge is not an accurate depiction of the manner in which the regression is capturing in the uncertainty in the fit. The way it's shown it implies that the regression is constrained to have a fixed value at t=t0, which is not the case (unless you specifically performed such a fit, which I doubt). I suggest you leave the envelope off to avoid this potential for confusion (I don't see a more accurate but clear way to depict this uncertainty graphically). The caption will need to be updated to match.

Line 9: Actually isn't this "mixing ratio" rather than "concentration"? (sorry to be picky)

Line 10: Actually the dashed line doesn't look that "heavy" to me.

Line 18/19: "...are shown as a percentage deviation..." sounds better to me.

Line 19: Define "seasonal mean", is it three-monthly averages (DJF, MAM etc.) or monthly averages?

Line 22: "look at" -> "examine" sounds more scientific to me.

— Page 4

Line 7: Perhaps "effects" -> "cycles"?

Lines 23-26: Add "MLS" before "HCl" (line 23) and "N2O" (line 24) and then delete "from MLS measurements of each constituent."

Line 24: add "a" before "de seasonalized"?

— Page 5

Line 2: "determined by" -> "that due to"

Figure 4: As with figure 2, I suggest you remove the "flared" red shading (and update caption accordingly).

[Figure]

Line 30 - Page 6 Line 1. The point about the "raw" and "Trend with N2O fit" being similar at the higher altitudes is a good one and makes geophysical sense to me. However, this then exposes a weakness in the authors' arguments and methods, in that the N2O drift and surface N2O trend terms add significantly to the "final" result, moving it far from the "raw" original. If dynamical variability is indeed "relatively small" at these altitudes then why do these modifying terms get the same "weight" at these upper levels as they do lower down where dynamical variability is significant? There seems to be some kind of inconsistency here that needs thought.

— Page 6

Lines 5-13: This is the other place where it would be good to talk about the validity of assuming pre- and post-2013 N2O drifts are consistent.

Table 1 caption: Suggest that you delete "with 2-sigma uncertainties" on line 18 and instead say at the end of the caption something like: "All uncertainties are quoted at 2-sigma".

Lines 25-30: This is where some discussion of age-of-air and related issues would clearly go.

— Page 8

Lines 12-14: Again, this point is seemingly at odds with the "final" results for the higher altitudes.

Line 29: "kkm" typo.

Lines 34-41: Doesn't the age-of-air spectrum come into this issue too? In any case, it would be best to "show your working" as to how the -4.9% estimate is arrived at here.

---

## Author Comment (AC1) · 15 Mar 2018

The referee's comments are presented followed by our responses in *italic* script.

**Anonymous Referee #1**

This manuscript introduces a method aiming at accounting for (correcting for) the real variability observed in time series of stratospheric tracers. More specifically, simultaneous measurements of hydrogen chloride (HCl, the main reservoir of stratospheric chlorine) and nitrous oxide (a long-lived source of nitrogen) are used in conjunction, so that the variability of stratospheric N2O is used to remove the one of HCl, assuming they have the same origins, here circulation and transport. The ultimate aim is to determine HCl trends unaffected by atmospheric variability, useful for the verification of the effectiveness of the Montreal Protocol in the stratosphere.

It is claimed that this method is more appropriate than those using proxies for representing multiyear dynamical variabilities resulting from, e.g., the QBO, ENSO. In the present case, the implementation of the method to simultaneous measurements of HCl and N2O by the Aura/MLS instrument results in the determination of significant HCl decreases in the lower stratosphere over the 2004-2016 time period (about 13 years) which are in agreement with the evolution of total organic chlorine at the surface, when accounting for the mean age of stratospheric air. In contrast, direct HCl trends are mostly not significant over this decadal time scale.

One can foresee that the method presented in this study will be used in future trend evaluations, for HCl, but also for other target gases relevant to ozone depletion and recovery, when trying to reconcile tropospheric and stratospheric trends,. . . provided that simultaneous measurements of tracers with similar response to atmospheric dynamical variability or other influences are available.

Therefore, I recommend publication of this study, after consideration of the suggestions indicated below.

**Major comments**

In order to allow precise implementations of this approach or replication of the method in future studies, it would be good to have available a description on how the "time-series regression" is actually performed (by simple ratioing or more elaborated ways). I believe this would not be possible with the current version of the text.

*See comment after next paragraph.*

Another aspect which is not described is the evaluation of the uncertainties affecting the various trends. It is stated in caption of Fig. 2 that "the 2σ uncertainty in that fit includes consideration of auto-correlation in the time series". Various approaches have been used in recent papers such as to account for auto-correlation in the data sets, particularly for studies of ozone recovery. But we do not know how the uncertainty ranges were determined in the present instance, while visual inspection of the HCl and N2O time series suggests that auto-correlation might be quite significant. It would be good to know how the authors accounted for auto-correlation. A brief description of the statistical evaluation of the confidence intervals should be added. A citation might be relevant if the current method has been used before.

*Have added a short paragraph explaining the time series model and the estimate of trend uncertainty including an estimate for the increase in uncertainty due to auto-correlation of the residuals. We have added a citation to Weatherhead et al. [1998] where the method is described in detail.*

**Specific comments and remarks**

There is a mismatch between the main text and the captions for Fig. 2 and 3. It is indicated in the text that anomalies or deviations are shown. But it looks like the captions correctly state that deseasonalized time series are shown for HCl and N2O. Several portions of the main body text need to be amended accordingly, or the captions and figures 2 and 3 updated.

*Changed "anomalies" to "time series" in the text*

**Minor comments or typos**

-Page 1/line 30: suggest changing to "... a few years for them to reach. . ."
*We prefer to spelling out "CFCs" rather than "them"*

-Page 1/line 37: Jungfraujoch is misspelt
*fixed*

-Page 1/line 38: suggest changing to ". . .from in situ surface measurements. . ."
*done*

-Page 2/line 14: might be good to indicate why N2O is a relevant target for this purpose (and/or add a reference)
*We feel that this paper explains why $N_2O$ is a relevant target. This is particularly evident in Figure 3 where we show the covariance of the time series of $N_2O$ and HCl. This is one of the main points of the paper.*

-Page 3/line 1: I don't think that there is a comparison between the N2O products from MLS, only the drift as a function of altitude is given
*This has been reworded to avoid the ambiguity..*

-Page 4/line 24: "deseasonalized" is misspelt
*fixed*

-Page 5/line3-line5: this sentence needs to be reworded
*Has been reworded to to indicate that stratospheric chlorine is expected to have changed due to changes in chlorine-containing source gases.*

-Page 5/line 16: "HCl anomaly time series as in Figures 2 and 3"; true if Fig. 2 and 3 are updated accordingly
*Reworded to indicate that it is same as Figures 2 and 3 with the mean removed.*

-Page 6/line28-30: a good reference is needed here, to introduce the concept of age of air, and showing that a 3 years delay is appropriate
*Added a reference to the review paper by Waugh and Hall. We do not feel the need to explain age of air as it is explained in detail in Waugh and Hall and is only a minor point in the present paper.*

-Page 7/figure 5: perhaps provide approximate altitude information on the right scale?
*Added words in caption to indicate approximate altitude.*

-Page 7/line 23: suggest replacing "showed" by 'reported"
*Changed*

-Page 7/line 24: suggest replacing "linear trend" by "linear decrease"
*Changed*

-Page 8/line 36: [Carpenter and Reimann, 2014] instead of 2015
*Fixed*

-Page 9/line 8: 2014 instead of 2015
*Fixed*

**Anonymous Referee #2**

This paper presents a nice perspective on observed variability and trends in northern mid-latitude stratospheric HCl. It describes an approach whereby dynamical influences on that variability can be accounted for through consideration of a trace gas such as N2O, which shares many of the dynamical influences as HCl but experiences different chemical processes.

While I recognize the value of the method described, and am keen to see it appear in the literature, I am concerned the uncertainties ascribed to some of the numbers found/used by the authors are on the optimistic side. I wonder if more complete assessment of these uncertainties might lead to a reduction in the reported "significance" of the result and this assessment might then suggest that a softening of some of the wording is merited. I also have a concern as to whether the ordering of the operations in their method is appropriate, and whether more robust results might be obtained if it were reversed. Both of these topics are expanded upon below.

The standard of English is reasonably high, but it would clearly have benefited from a more careful read through by the authors as there are several parts that are erroneously and/or ambiguously worded. I've endeavored to identify some of these, but fear I may have overlooked some others.

**Major concerns**

My concern about the ordering of steps in the method is as follows. I would have thought that it would have been better to "correct" the N2O for both the likely MLS drift and the surface growth rate before using it as an explanatory variable in the HCl analysis rather than, as appears to be the case, after. My sense is that this would lead to a corrected N2O variable that would do a better job of explaining the dynamical influences on the HCl, enabling a clearer trend to be obtained. The results may be little different in the end, but my sense is that the study would be better expressed in that manner. If nothing else, the authors would do well to enact that alternative formulation and comment on the difference it makes to the result (even if they chose not to show it in the end). It might make sense to include an actual algebraic expression for the fit and the various corrections. This would make for an easier description for the various terms involved and their uncertainties.

*Actually, to correct the MLS drift we fit the difference between the bands with a linear trend plus a seasonal cycle in the mean and a seasonal cycle in the trend. In the end, we simply took the linear trend portion of the fit to the difference between the two bands. We did not feel that the knowledge of the drift, and how to extrapolate it, was good enough to justify more than just a simple linear correction. Using the simple linear trend means that it makes no difference to the end result where in the process we apply the correction. The uncertainty in the fit trend was obtained in the same way as all of the linear trend uncertainty estimates in the paper, from the standard deviation multiplied by a factor from the Weatherhead et al. reference to account for autocorrelation of the residuals. We have added a short explanation of this reasoning.*

My more major concern relates to the uncertainties quoted for some of the results. This is particularly important given the extent to which many of them are only just statistically significant (using the authors' 2-sigma threshold). Firstly, it is clear that the level-to-level variations in the bottom line results are mostly driven by the reported N2O 190/640 drift (r=0.75 between it and the result) rather than by the observed HCl trend (r=-0.24). That is to say, the results are affected more by the "correction" than by the actual input (the latter being the HCl trend with the N2O fit term included). Accordingly, this correction deserves particular scrutiny. The degree of level-to-level changes in this drift term is large compared to the uncertainty quoted on many of the individual drifts. Arguably, the standard deviation (1-sigma=1.5%/decade) of these different estimates would be just as valid a measure of the uncertainty in any or all of them. Indeed it might have been just as valid to chose to use the multi-level-mean drift as the value for all levels, given the uncertainty introduced by the inherent assumptions being made. Foremost among those assumptions is the

one that the N2O drifts seen in the first part of the MLS mission are the same as those expected in the post-2013 period, when the 640 GHz N2O product is unavailable. I would have thought that the uncertainties derived here might need to be inflated in some way to account for this. Might more information be gained through consideration of other MLS products measured in the same period? Fundamentally, I think more information is needed here (including from the MLS team) on these uncertainties and their validity.

*We do appreciate this concern. Our goal was to emphasize the concept of modeling variability in measurement time series by using the variability of another measured constituent. We have added a statement to this effect in the abstract and a paragraph at the end of the paper to mention these concerns and emphasize the conceptual focus of the paper.*

My second concern on the uncertainty relates to the 0.05%/decade (2-sigma) uncertainty quoted on the impact of N2O emissions. Firstly, the use of a constant 2.8%/decade trend at all altitudes here strikes me as highly simplistic. There are factors such as changes in age of air (and its spectrum) that surely come into play and might lead to variations. Similarly, the use of a 3-year lag at all altitudes seems overly simplistic. I grant that these issues may only have a small impact, and they may be very hard to quantify from the measurements available. Thus, the use of a constant value may well be justified in that light. However, I find it hard to believe that, in the face of those issues, the 0.05%/decade 2-sigma uncertainty estimate is an appropriate one.

*Since the surface $N_2O$ trend is close to linear, the time delay has little impact on the results and contributes little to the uncertainty.*

If nothing else, I would urge the authors to validate this number through, for example, examination of CCM runs (to which this team has ready access). Quantifying the degree to which the modeled 45N N2O timeseries at different pressure levels tracks the surface trend would provide a useful measure of this uncertainty.
*Actually we first realized the possibility of this approach from model results. The model has better correlations than the data. We feel that the measurements shown in Figure 3 with the high degree of correlation between HCl and $N_2O$ clearly illustrate that these quantities are correlated in the atmosphere. Citing details from model results would not add much to the discussion.*

This issue is perhaps tied up with the ordering one discussed above, as the use of N2O as an explanatory variable for the sought-after HCl trends may absorb these factors to some extent (though I haven't thought this through fully). In my mind all these issues argue that a more complete exploration of their methods, their inherent assumptions, and the uncertainties therein should be included in the manuscript.

*We understand that this is simplistic. The idea was to keep from overcomplicating the analysis to keep from obscuring the main point about the concept of using measured constituent variability in place of standard proxies for dynamical variability. We have added sentences in the abstract and in the conclusion that broaden the point a bit by pointing out that even if you do not use the second constituent ($N_2O$ in this case) as a direct proxy, you do gain important information by examining the time series of other species that co-vary with the one you are considering (HCl in this case). This information is particularly important when dynamics may be the cause of the apparent "trend". Strahan et al. (2011, JGR) showed that $N_2O$ and mean age have a linear releationship up to 30 hPa in the midlatitudes. This means that age spectrum variations are not important to the midlatitude $N_2O$ used in the 32-68 hPa range of this study. That paper also showed mean ages of 2-4 years in this region, which is why a 3-year lag between the surface and the lower stratosphere was chosen.*

**More minor points**
— Page 1

Line 19: "Statistically" -> "Statistical"
*fixed*

Line 25: "altitude" -> "vertical" (as you're using pressures rather than altitudes in words that follow).
*Changed to "vertical pressure"*

Line 27: "... amount of inorganic stratospheric chlorine. This marker can be ..." to avoid the ambiguity about whether it is the HCl or the inorganic chlorine that "can be measured from the ground and from satellites".
*fixed*

Line 33: Commas needed after "showed" and "measurements"
*fixed*

Line 34: "Inorganic chlorine" is more than just HCl and ClONO2, though granted the others may be minor. Or is the point that Rinsland et al. only measured those two species and argued that they are the bulk inorganic chlorine. Please clarify.
*added clarification*

Line 37: Jungfraujoch misspelt
*fixed*

Line 38: "during the early 2000s. This was followed by an increase in the HCl column over Jungfraujoch from ..." to avoid the ambiguity about whether it is the HCl or the source gases (the most recent things being discussed) being referred to.
*clarified*

Lines 39-43: The way this is worded, it seemingly ignores the fact that Mahieu et al. also looked at this signal in MLS data (as embodied in the GOZCARDS dataset). Please reword accordingly.
*added clarification*

— Page 2

Line 3: Quote the latitude of Jungfraujoch in the caption.
*done*
Also, some redundancy, as you say the MLS data is a 100-10hPa column in one sentence and then talk about it being a partial column (without the numbers) later on.
Lines 10-15: Again, please be sure your wording is consistent with the use to which Mahieu et al. put MLS data.
Line 10: "results from simulations using the SLIMCAT model driven by..."
*fixed*

Line 21: July 2004 doesn't sound like "late 2004" to me.
*changed to July*

Line 23: "altitude" -> "vertically resolved", given that the vertical coordinate is pressure.
*changed to "vertical"*

Line 26: "has little change since" -> "shows little change from"
*fixed*

Line 30: Perhaps put "band 14" in quotes as it's jargon that's not explained earlier (and is presumably covered in the references given earlier in the paragraph).
*removed*

Lines 32-34: Please clarify, has the N2O product been "redefined" since the release of v4.2, or was the redefinition part of v4.2 from the outset?
*We have replaced the paragraph describing the N2O product with a new paragraph that more explicitely explains the situation.*

Line 34: Unless I've misunderstood, it's part of MLS that has "deteriorated" is it not? Starting at some point during the mission. The way this is worded it sounds like the MLS data files are somehow deteriorating with time (like food going off in the refrigerator) regardless of the time at which the observations were made. Please reword more precisely. —
*replaced entire paragraph with what we hope is a clearer discussion*

Line 1: "next" -> "following" sounds better to me. Line 2: Are the "640 channel" measurements also from the v4.2 dataset or from some earlier version?
*As stated above we have rewritten the description to be more explicit about the data products.*

Lines 1-6: This would presumably be a good place to have a discussion about the validity of assuming that the pre-2013 drifts are representative of the post-2013 observations. (Or possibly on page 6, see later).
*Added a description later in the discussion of Table 1.*

Figure 2: The way you've drawn this, with the shaded envelope being narrow at the left hand edge is not an accurate depiction of the manner in which the regression is capturing in the uncertainty in the fit. The way it's shown it implies that the regression is constrained to have a fixed value at t=t0, which is not the case (unless you specifically performed such a fit, which I doubt). I suggest you leave the envelope off to avoid this potential for confusion (I don't see a more accurate but clear way to depict this uncertainty graphically). The caption will need to be updated to match.

*Thanks for pointing this out. We have replaced Figures 2 and 4 with new versions showing the shaded area coming to a point through the middle. This better represents the actual meaning of the regression uncertainty.*

Line 9: Actually isn't this "mixing ratio" rather than "concentration"? (sorry to be picky)
*fixed*

Line 10: Actually the dashed line doesn't look that "heavy" to me.
*Agreed and fixed*

Line 18/19: "...are shown as a percentage deviation..." sounds better to me.
*fixed*

Line 19: Define "seasonal mean", is it three-monthly averages (DJF, MAM etc.) or monthly averages?
*Improved the description to describe that we removed the seasonal cycle, while retaining the mean and have plotted the percent deviation of the residual from the mean.*

Line 22: "look at" -> "examine" sounds more scientific to me.
*fixed*

— Page 4
Line 7: Perhaps "effects" -> "cycles"?
*Decided to stay with "effects". We think of the seasonal effect as a cycle. The QBO is an oscillation having an irregular frequency, especially for the last few years.*

Lines 23-26: Add "MLS" before "HCl" (line 23) and "N2O" (line 24) and then delete "from MLS measurements of each constituent."
*Changed and added "measurements" after each.*

Line 24: add "a" before "deseasonalized"?
*fixed*

— Page 5

Line 2: "determined by" -> "that due to" Figure 4: As with figure 2, I suggest you remove the "flared" red shading (and update caption accordingly).
*Also replaced Figure 4 same change as with Figure 2.*

- Page 6 Line 1. The point about the "raw" and "Trend with N2O fit" being similar at the higher altitudes is a good one and makes geophysical sense to me. However, this then exposes a weakness in the authors' arguments and methods, in that the N2O drift and surface N2O trend terms add significantly to the "final" result, moving it far from the "raw" original. If dynamical variability is indeed "relatively small" at these altitudes then why do these modifying terms get the same "weight" at these upper levels as they do lower down where dynamical variability is significant? There seems to be some kind of inconsistency here that needs thought.
*The 'trend with $N_2O$ fit' and drift rate only have significant effects on the net trend at pressures 22-68 hPa. At altitudes above 22 hPa, Cly is nearing its maximum stratospheric value and thus becomes insensitive to dynamical variability. Note that at 10 and 15 hPa the 'trend with $N_2O$ fit' is balanced by the surface trend – an indication of the reduced sensitivity to $N_2O$ (dynamics) here.*

— Page 6
Lines 5-13: This is the other place where it would be good to talk about the validity of assuming pre- and post-2013 N2O drifts are consistent.
*We have added some clarifying discussion of this problem here and in the conclusions.*

Table 1 caption: Suggest that you delete "with 2-sigma uncertainties" on line 18 and instead say at the end of the caption something like: "All uncertainties are quoted at 2-sigma".
*done*

Lines 25-30: This is where some discussion of age-of-air and related issues would clearly go.
*This paragraph has been rewritten to be clearer and we have added a basic reference to age-of-air (Waugh and Hall). The age-of-air issue is extremely minor in this case.*

— Page 8
Lines 12-14: Again, this point is seemingly at odds with the "final" results for the higher altitudes.
*This statement refers to the effect of using the $N_2O$ time series as a proxy for dynamical variability. The raw trend and corrected trend in columns 1 and 2 of Table 1 are approximately equal. The corrections do lead to a difference in the "final" results.*

Line 29: "kkm" typo.
*fixed*

Lines 34-41: Doesn't the age-of-air spectrum come into this issue too? In any case, it would be best to "show your working" as to how the -4.9% estimate is arrived at here.
*Considering the age-of-air spectrum here would be over complicating this simple estimate of the slope of chlorine expected during this time period. We have changed the statement to an estimate of -5%/decade because the original 4.9%/decade overstated the significant figures. Our main point here is that we obtained a reasonable result by applying the $N_2O$ as a proxy.*